# Time-resolved chemically-selective spectroscopic investigation of the redox reaction between hematite and aluminium

Ettore Paltanin [1,2,6], Jacopo S. Pelli Cresi[1,6], Emiliano Principi [1] ✉, Wonseok Lee [3], Filippo Bencivenga[1], Dario De Angelis [1,4], Laura Foglia [1], David Garzella[1], Gabor Kurdi [1], Michele Manfredda[1], Denys Naumenko[1,5], Alberto Simoncig[1], Scott K. Cushing[3], Riccardo Mincigrucci[1] ✉ & Claudio Masciovecchio [1]

Thermite reactions –highly energetic redox processes between a metal and an oxide–are used in welding, propulsion, and the fabrication of advanced materials. When reduced to the nanoscale, these reactions exhibit enhanced energetic performance, but their ultrafast dynamics remain poorly understood. Gaining insight into charge transfer during these processes is essential for advancing applications in energy conversion and materials design. Here we show that the reaction between aluminium and hematite, a common iron oxide, can be tracked with femtosecond resolution using extreme ultraviolet (EUV) time-resolved absorption spectroscopy at the Fe $M_{2,3}$ and Al $L_{2,3}$ edges. By exciting the system with an ultrashort optical pulse and probing element-specific absorption changes, we observe an early spectral shift that reveals the formation of localized charge carriers (polarons). Comparing samples with different supporting substrates highlights ultrafast electron transfer from aluminium to hematite. These results demonstrate an approach to investigating charge flow in energetic materials and provide a basis for studying fast chemical reactions with chemical specificity.

Redox reactions are fundamental chemical processes involved in many vital systems, such as cellular respiration and photosynthesis, as well as in combustion and fermentation. Electron transfer is the initial and fastest step of a redox reaction, typically occurring within hundreds of femtoseconds. The rapid redistribution of electrons initiates bond breaking and formation, followed by nuclear rearrangement on the picosecond timescale. Thermite reactions–redox processes between a metal and an oxide–are particularly exothermic and irreversible. They are widely used in welding, materials synthesis, propulsion and pyrotechnics. When at least one of the reactants is nanostructured, these systems are referred to as metastable intermolecular composites

(MICs), which exhibit enhanced performance due to lower ignition thresholds and faster reaction kinetics enabled by increased surface contact[1,2]. The interest in MICs has led to studies using differential scanning calorimetry[2], X-ray diffraction[3,4], Mössbauer spectroscopy[3], mass spectrometry[5], and photoemission spectroscopy[6], under various ignition conditions including electrical discharge[3], furnace heating[4], and nanosecond pulsed lasers[1,2,5].

These prior investigations focused on microsecond-to-second timescales, targeting slower steps like oxygen migration and complete product formation. In contrast, the ignition phase itself typically unfolds over nanoseconds or longer, requiring higher temporal

[1]Elettra-Sincrotrone Trieste S.C.p.A., 34149Basovizza, Trieste, Italy. [2]Dipartimento di Fisica, Università degli Studi di Trieste, 34127 Trieste, Italy. [3]Division of Chemistry and Chemical Engineering, California Institute of Technology, 91125 Pasadena, CA, USA. [4]CNR - Istituto Officina dei Materiali (IOM), AREA Science Park, Basovizza, 34149 Trieste, Italy. [5]Infineon Technologies, 9500 Villach, Carinthia, Austria. [6]These authors contributed equally: Ettore Paltanin, Jacopo S. Pelli Cresi. ✉e-mail: emiliano.principi@elettra.eu; riccardo.mincigrucci@dynamic-optics.it

resolution to be properly investigated. Ultrafast soft X-ray and EUV spectroscopic techniques have been effectively applied to intramolecular photochemical reactions[7,8], but chemically resolved studies of bimolecular reactions on the femtosecond scale remain absent. Time-resolved X-ray absorption spectroscopy (tr-XAS) at the Fe $M_{2,3}$ edge, sensitive to oxidation state changes in $\alpha$-Fe$_2$O$_3$, has proven to be a powerful probe of ultrafast dynamics[9,10].

In this work, we investigate the redox reaction between aluminium and hematite using femtosecond EUV transient absorption spectroscopy. By initiating the process with an ultrashort optical pulse and probing the dynamics at the Fe $M_{2,3}$ and Al $L_{2,3}$ absorption edges, we track early electron transfer events with chemical selectivity (Fig. 1). This approach enables the first femtosecond-resolved, element-specific observation of a bimolecular redox reaction involving a thermite system at the nanoscale.

## Results

### Pump-probe dynamics

The transient transmission is calculated as follows:

$$\Delta T = T_{pp} - T_{pre} \tag{1}$$

where $T_{pre}$ is the transmission of the unperturbed sample, and $T_{pp}$ is the transmission after pump excitation. In two different sets of measurements, dynamics were recorded for several photon energies across the Al $L_{2,3}$ edge and Fe $M_{2,3}$ edge for the $\alpha$-Fe$_2$O$_3$/Al sample (see Fig. 2a, 2c). The temporal sampling has been chosen in such a way as to probe both ultrafast electron-phonon coupling dynamics in the vicinity of time zero with steps of 100 fs and the slower lattice dynamics at larger delays with steps of 10 ps, while intermediate steps of 1 ps in the region of transition between these two regimes are also

taken. The pump-probe traces are reported in Fig. 2b, d along with their fits. The functions employed for the fits are described in the SI. The time constants obtained from the fitting procedure are listed in Table 1.

The time constants for the fast part of the dynamic, i.e. $\tau_1$, derived from the fit result in few picoseconds for both $\alpha$-Fe$_2$O$_3$ and Al. For Al, $\tau_1$ is reasonable for the equilibration of hot electrons with the lattice via electron-phonon collision in a metal whose behavior resembles that of an ideal free-electron gas and could be described with the two temperature model (TTM)[11,12]. For $\alpha$-Fe$_2$O$_3$, $\tau_1$ is larger than for Al and this could be ascribed to the different relaxation dynamics. In the case of hematite, excited carriers interact with the lattice via scattering off electrons with optical phonons which leads to small polaron formation and localization of the charge carrier[13]. The dynamics at 56.49 eV required a second exponential decay ($\tau_1^* = 0.1 \pm 0.06$ ps) to fit the data (see SI). The presence of this additional term is justified by the fact that this specific photon energy is close to the spectral region where the feature associated to the process of small polaron formation is observed (namely, a blue shift of the invariant absorption point, which is highlighted in Fig. 3 and better described later in the text). The

**Table 1 | Kinetic constants from the fits of the pump-probe traces for aluminium and hematite**

|  |  | Fast | Slow |
|---|---|---|---|
|  |  | $\tau_1$ (ps) | $\tau_2$ (ps) |
| Pre-edge | Al | 1.18 ± 0.13 | 8.3 ± 1.1 |
|  | Fe | 2.60 ± 0.08 | 42.0 ± 10.9 |
| Post-edge | Al | 1.54 ± 0.21 | 3.3 ± 0.3 |
|  | Fe | 1.58 ± 0.16 | 40.0 ± 6.9 |

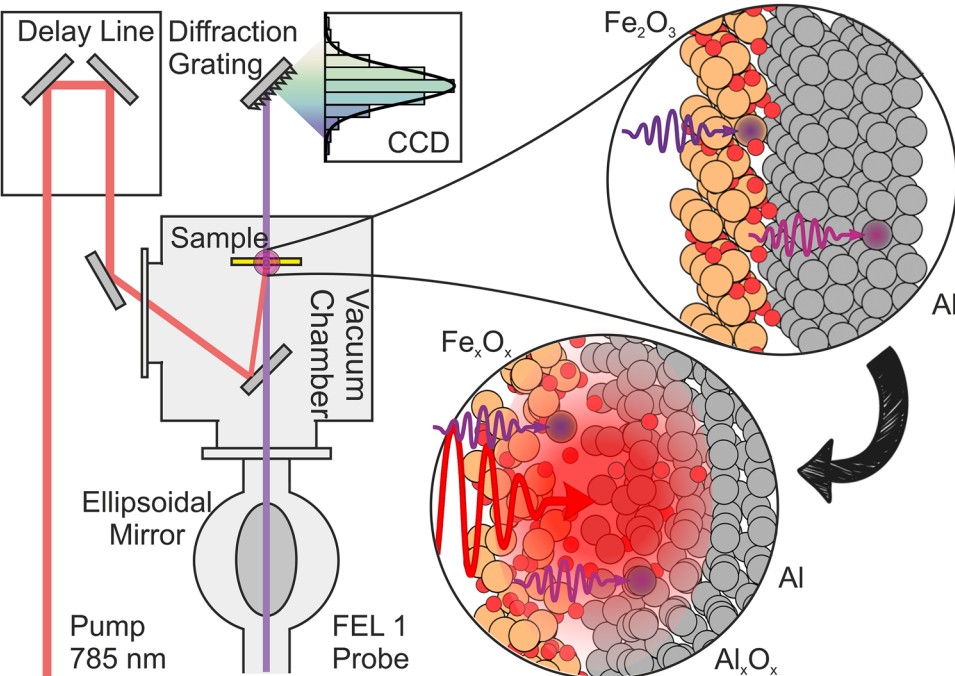

**Fig. 1 | Sketch of the experimental layout.** The red line represents the path of the visible pump laser, which is delivered in the experimental chamber (in high vacuum) from the femtosecond laser facility of FERMI and it is focused on the sample using a lens. The free electron laser (FEL) radiation is employed to probe the dynamics at the Fe $M_{2,3}$ and Al $L_{2,3}$ absorption egdes and it is illustrated by the violet line. It is delivered to the sample chamber via the beam transport vacuum pipes and it is focused on the sample with an ellipsoidal mirror. A motorized hollow mirror allows to work with the laser and the FEL beams in a quasi-collinear geometry at the sample. The transmitted extreme ultraviolet (EUV) radiation is detected with a spectrometer composed by a diffraction grating and CCD camera, while the pump is filtered. The laser pump drives the sample out of equilibrium, reaching high temperatures and triggering the redox process between $\alpha$-Fe$_2$O$_3$ and Al. The label "FEL 1" refers to the FERMI EUV source operating in the low-photon-energy range[23]. In the sketches, the orange spheres represent Fe atoms, the red spheres represent the O atoms and the gray spheres represents Al atoms.

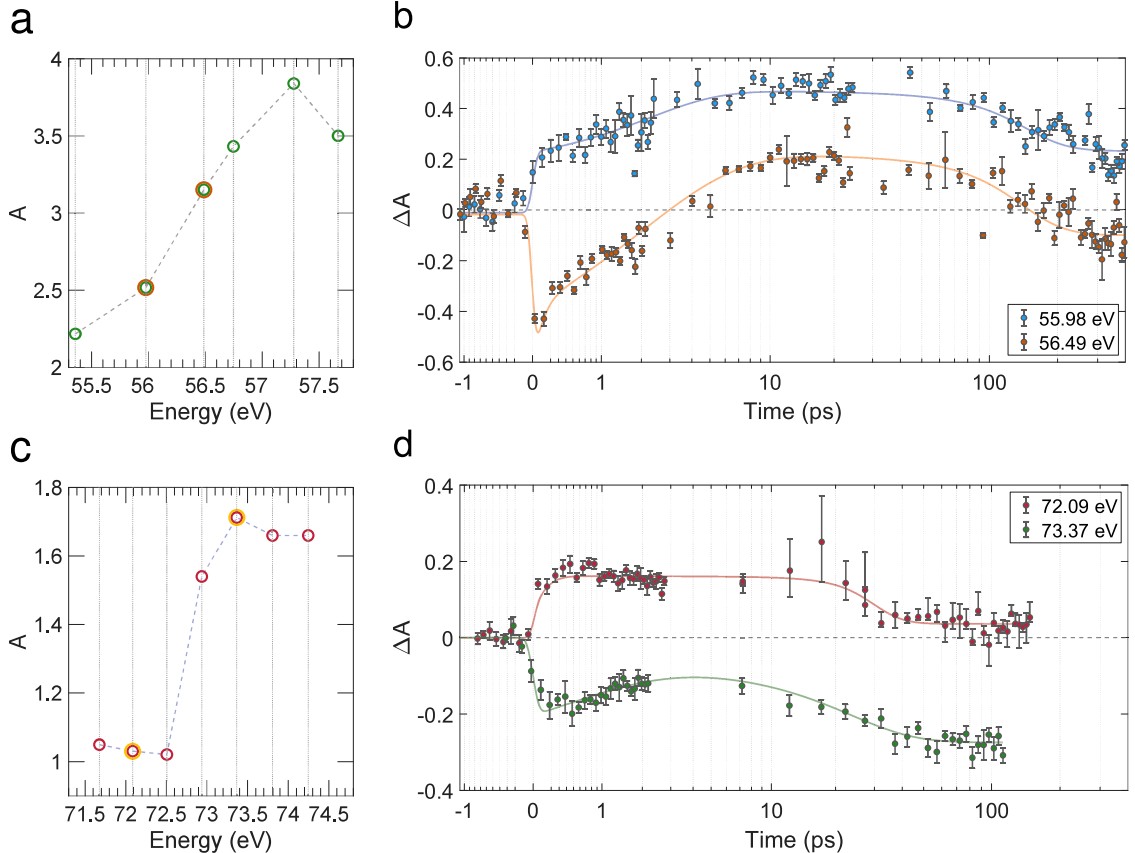

**Fig. 2 | Static absorption spectra.** Absorption at the $M_{2,3}$ edge of iron in hematite (**a**) and $L_{2,3}$ edge of aluminium (**c**) measured on the $\alpha$-$Fe_2O_3$/Al sample. Absorption dynamics probed at 55.98 eV and 56.49 eV for the Fe $M_{2,3}$ edge (**b**) and 72.09 eV and 73.37 eV Al $L_{2,3}$ edge (**d**). The photon energies which were investigated with the FERMI FEL are highlighted in green in panel (**a**) and in red in panel (**c**). The photon energies whose dynamics are shown in the panel (**b**) are highlighted in orange in panel (**a**) and the photon energies whose dynamics are shown in the panel (**d**) are highlighted in yellow in panel (**c**). The functional form of the curves in **b** and **d** is provided in the Supplementary Information, Eqs. 8,9,10. Parameters of the used curves: $f$(55.98 eV): $A_1 = 0.4825$, $A_2 = 0.2498$, $\tau_1 = 1.8$, $\tau_2 = 40$, $z = 122$, $\sigma = 0.08$; $f$(56.49 eV): $A_1 = 0.5955$, $A_1^* = -0.2597$, $A_2 = 0.3397$, $\tau_1 = 3.0$, $\tau_1^* = 0.1$, $\tau_2 = 40$, $z_2 = 115$, $\sigma = 0.07$; $f$(72.09 eV): $A_1 = 0.1622$, $A_2 = -0.1261$, $\tau_1 = 0.1$, $\tau_2 = 5.45$, $z = 27.8$, $\sigma = 0.08$; $f$(73.37 eV): $A_1 = -0.1276$, $A_2 = -0.2746$, $\tau_1 = 1.45$, $\tau_2 = 11.76$, $z = 11.3$, $\sigma = 0.1$. Error bars are calculated using Eq. 5 in the Supplementary Information. Source data are provided as a Source Data file.

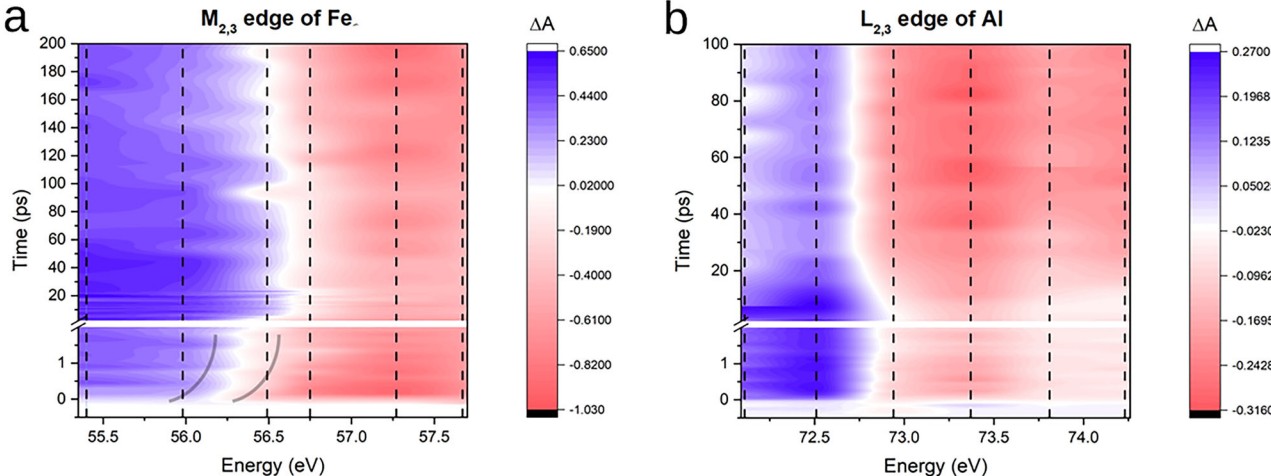

**Fig. 3 | Transient absorption traces of the $\alpha$-$Fe_2O_3$/Al sample.** Transient absorption (TA) traces of the $\alpha$-$Fe_2O_3$/Al sample across the Fe $M_{2,3}$ (**a**) and Al $L_{2,3}$ (**b**) absorption edges. The dashed lines are the pump-probe dynamics measured in the experiment and interpolated over a uniform time mesh in order to plot them together. The 2D-maps are obtained by interpolation of the time traces over the photon energies measured in the experiment and serves the purpose of providing an overall picture of the results of the dynamic behavior of the absorption edges. Source data are provided as a Source Data file.

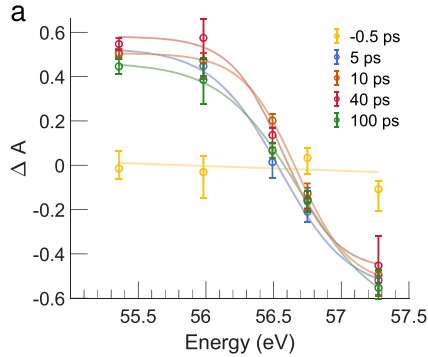
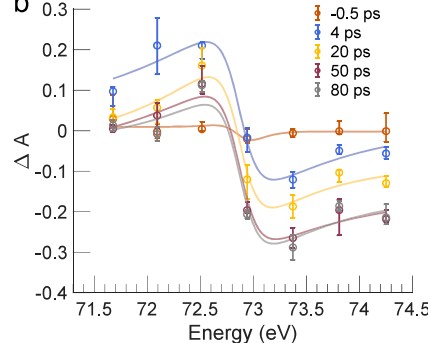

**Fig. 4 | Transient absorption spectra of the $\alpha$-Fe$_2$O$_3$/Al sample.** Transient absorption (TA) spectra at some representative delays for Fe M$_{2,3}$-edge of hematite (**a**) and Al L$_{2,3}$-edge of aluminium (**b**). The flexible sigmoid function used for the fits is reported in the SI. For the Al TA, the difference between two sigmoid functions has been employed for the fitting. Error bars are calculated using Eq. 5 in the Supplementary Information. Source data are provided as a Source Data file.

polaron is interpreted as an additional decay because its formation is associated to the disappearance of hot electrons from the conduction band. The time constant for the second part of the dynamic, i.e. $\tau_2$, is on the order of tens of picoseconds for aluminium and on the order of tens to hundreds of picoseconds for hematite. Several studies on fs laser ablation have been performed on aluminium with different fluence regimes[14–17]. High fluences (> 0.05 J/cm$^2$) have been demonstrated to provoke ultrafast isochoric heating, which leads to the collapse of the overheated material and to the generation of a compression pressure of GPas in the central part of the film. Under such conditions, the film is forced to expand, forming low density regions (voids) and nanoparticles within the first tens of picoseconds from the impinging of the laser pump. The time constants $\tau_2$ reported in Table 1 and the uniform increase of the transmission at longer delays are in good agreement with the mechanism of ultrafast isochoric heating just described. The absolute variation of the transmission in Al could be associated with a reduction of the number of atoms present in the volume probed by the FEL of ~ 8%. We did not find similar studies of fs laser ablation on hematite, but since our pump-probe traces have the very same outline for aluminium and hematite, just with different kinetic constants, we can conclude that a similar mechanism of isochoric heating is occurring in hematite as well with high laser fluences. A similar estimation on the reduction of the Fe atoms in the probed volume has been performed, resulting in a decrease of ~ 5%. Nevertheless, the variation of the absorption cross-section is likely to depend on other phenomena as well, *i.e.* phase transformations and change of the refractive index. The significantly higher kinetic constants in the case of hematite (a wide-band gap semiconductor) can be explained in terms of a lower heat conductivity and a higher heat capacity. As a consequence, the heating is smaller and the consequent expansion of the lattice is slower.

Understanding the dynamics of the two species that compose our bilayer sample was essential for establishing on a solid ground whether or not the requirements for starting the thermite redox process are meet in our experiment. Studies on the thermite reaction[3,5,6] have identified the melting of aluminium as a fundamental step for triggering the redox reaction between aluminium and iron oxide. In the simulation from Tang et al.[14], a 200 fs laser pulse with a fluence of $\simeq$ 0.06 J/cm$^2$ provokes thermal melting via collapse of the crystal structure of a 100 nm thick aluminium film within the first 2 ps, raising the temperature of the lattice on the side of the sample exposed to the beam up to 3000 K. In our experiment, the laser pump fluence was set to $\simeq$ 0.9 J/cm$^2$, therefore the temperature of the lattice on the side of the film facing hematite has been raised at temperatures even higher than 3000 K, ensuring that the key condition for triggering the redox process, i.e. melting of aluminium, is satisfied. Furthermore, the collapse of the

crystal structure, the expansion/disgregation of the individual layers forced by the compressive pressure and the formation of nanoparticles seems to be a condition that favors the mixing and maximizes the contact area between the two species at the interface.

In Fig. 3 the pump-probe traces measured at several energies for $\alpha$-Fe$_2$O$_3$ (left panel) and Al (right panel) are plotted together to provide a comprehensive picture of the results of the measurement and to help visualizing the trend of the dynamic traces across the two absorption edges. The sign of the variation of absorbance reflects the broadening of the Fermi-Dirac distribution caused by the raise of the temperature of the electrons upon excitation by the laser pump. Indeed, the blue regions at lower energies are associated with an increase of absorbance while the red regions at higher energies are associated with a decrease of absorbance. Figure 4 shows static spectra and transient spectra for different pump-probe delays that help in visualizing the trend of variation of the absorption. In the two TA maps, the local variation of the density of the two materials, that are forced to expand by the compressive pressure which arises from the ultrafast isochoric heating, can also be observed in the decrease of absorbance that consistently occurs after tens of picoseconds (the blue region becomes weaker, the red region becomes more intense in both the TA maps).

## Transient absorption spectra at the Fe M$_{2,3}$ edge

In the study of Carneiro et al.[10] it has been demonstrated that a specific spectroscopic feature in the TA spectrum, i.e. the shift of the zero-crossing in the first hundreds of fs, is the fingerprint of small polaron formation in the spectral region between 56 and 58 eV. Polaron formation is a two-step process: (i) the first step is the optical excitation, which has a charge transfer nature, involving the displacement of an electron from the O 2p orbitals to the Fe 3d orbitals and, consequently, a variation of the oxidation state of Fe from +3 to +2; (ii) the second step is the interaction of the carriers with optical phonons provoking a local distortion of the lattice, which induces self-trapping. They propose a kinetic model where polarons are formed by bimolecular recombination of hot electrons and optical phonons. Polarons appear within 100 fs and their population continues to grow for 2-3 ps, when the hot electrons and phonons populations are consumed. Comparing our TA measurement at Fe M$_{2,3}$ edge with the study of Carneiro et al., we have observed the same feature, regardless the fact that we employed a 785 nm pump, which is well below the $\alpha$-Fe$_2$O$_3$ bandgap ($\simeq$ 560 nm), while they were using an optical pump above the bandgap of $\alpha$-Fe$_2$O$_3$. The left panel of Fig. 3 shows a detail of the transient absorption spectrum of hematite (on aluminium) at the Fe M$_{2,3}$ edge in the first picoseconds of the pump-probe measurement. The shift of the zero crossing point[10] is highlighted by the gray lines. Such observation

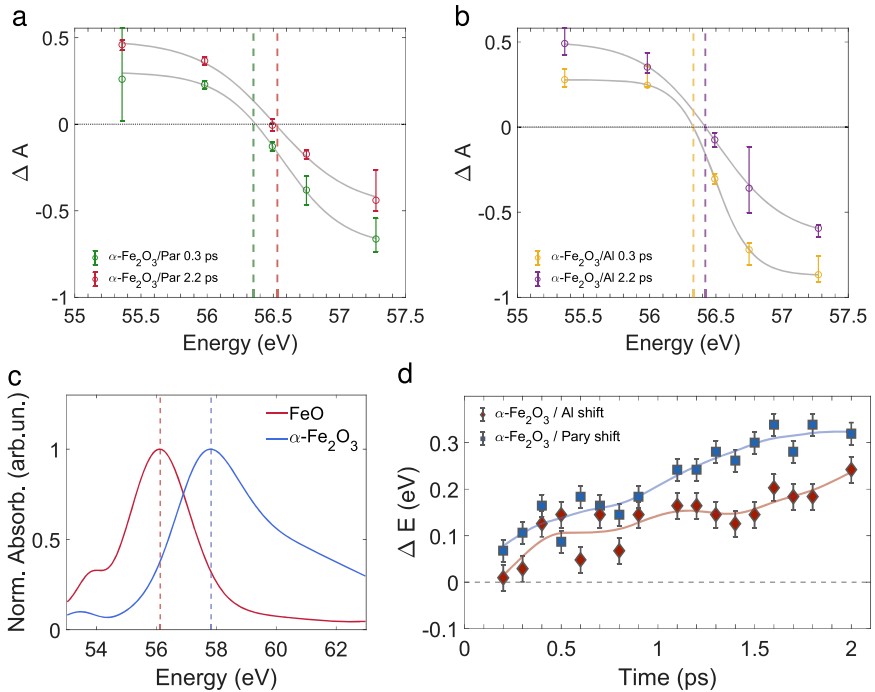

**Fig. 5 | Comparison of the shift in the zero-crossing point between $\alpha$-Fe$_2$O$_3$ on Al and $\alpha$-Fe$_2$O$_3$ on parylene.** Spectra from the TA trace of the $\alpha$-Fe$_2$O$_3$/parylene sample (**a**) and of the $\alpha$-Fe$_2$O$_3$/Al sample (**b**) at the Fe M$_{2,3}$ absorption edge. Two delays were selected (0.3 and 2.2 ps) to show the shift of the zero-crossing. **c** Simulation of the static absorption spectra for the Fe M$_{2,3}$-edge for FeO and $\alpha$-Fe$_2$O$_3$. **d** Evolution of the shift of the zero crossing point with respect to the crossing at $t = 0.1$ ps for the $\alpha$-Fe$_2$O$_3$/Al sample (red) and for the $\alpha$-Fe$_2$O$_3$/parylene sample (blue). The errors are calculated as the root means square of the residuals with respect to the fit of the two distributions. Source data are provided as a Source Data file.

poses a question on the nature of the excitation mechanism. According to the literature, two-photon absorption (TPA) is the dominant mechanism for excitation in hematite for photon energies below its bandgap. Okazaki et al.[18], demonstrated this by studying the excitation mechanism of a hematite photoanode in contact with a gold nanorod using transient absorption spectroscopy. Their findings showed that TPA must be considered to explain the dynamics of hematite transient absorption spectra when using an excitation wavelength of 800 nm. Additionally, good efficiencies for non-linear optical processes in the near-infrared region, such as at 1064 and 1340 nm[19], further support the occurrence of TPA in hematite at longer wavelengths relative to its visible bandgap. Since TPA in hematite was observed using a laser pump at 800 nm with a fluence of 1.0 mJ/cm$^2$, in the case of our experiment TPA is most likely the dominant mechanism of excitation.

### Comparison of the shift in the zero-crossing point between $\alpha$-Fe$_2$O$_3$ on Al and $\alpha$-Fe$_2$O$_3$ on parylene

In Fig. 5, transient soft-X-ray absorption spectra at the Fe M$_{2,3}$ edge are reported for the $\alpha$-Fe$_2$O$_3$/Parylene sample and for the $\alpha$-Fe$_2$O$_3$/Al sample. The spectra are fitted over the transient absorption values from the single wavelength pump probe experiments across the Fe M$_{2,3}$ edge for two selected delays. The fit is weighted on the errors of the individual experimental points and the details of the fitting procedure are described in the SI. The difference of the spectral behavior in time is highlighted in Fig. 5a and 5b by the dashed vertical lines, that help to visualize the shift of the zero-crossing point with the pump-probe delay. Comparing our spectra of hematite deposited on Al and on parylene, we have observed a noticeable difference in the blue shift of the zero-crossing point, that we quantified to be ~ 0.1 eV for $\alpha$-Fe$_2$O$_3$ on Al and ~ 0.2 eV for $\alpha$-Fe$_2$O$_3$ on parylene going from 0.3 ps to 2.2 ps. To confirm this observation, we computed the shift for several delays, taking as a reference 52.2 eV, the value of the zero-crossing at 0.1 ps. We monitored the behavior of the shift for the $\alpha$-Fe$_2$O$_3$/Al sample and

for the $\alpha$-Fe$_2$O$_3$/parylene sample. The trend observed comparing Fig. 5a and 5b is consistent for several delays and it is shown in Fig. 5d. The $\alpha$-Fe$_2$O$_3$/Al sample shows a smaller blue shift. The origin of the smaller blue shift observed in the case of the $\alpha$-Fe$_2$O$_3$/Al sample could be attributed to the injection of charge from Al to $\alpha$-Fe$_2$O$_3$. The electrons injected to $\alpha$-Fe$_2$O$_3$ are likely to be localized in the form of small polarons as well, but the mechanism might be different because the charge balance of the material is overall altered. This interpretation is supported by the observation of a reduction of $\alpha$-Fe$_2$O$_3$ by a metal in the aforementioned study from Okazaki et al.[18]. Furthermore, it is reasonable to think that this phenomenon would occur in the experimental conditions of the present investigation, since we proved to satisfy the conditions to start the reaction and the reduction of hematite is reported to be the first step of the thermite reaction[3]. Additionally, to corroborate our hypothesis, a red shift of the transient signal has been predicted from the calculation performed on hematite in the work from Klein et al.[20] when excited carriers are included in the computation of the polaron state. Finally, a red shift of the Fe M$_{2,3}$ absorption edge is predicted for a reduction of Fe from +3 to +2 oxidation state, as it can be seen from the result of the simulation of the static absorption of FeO and $\alpha$-Fe$_2$O$_3$ in Fig. 5c. Our results support the idea that an electron transfer between aluminium and hematite is taking place upon excitation of the $\alpha$-Fe$_2$O$_3$/Al sample with a high power 785 nm pulse. On longer time scales (tens to hundreds of picoseconds), we have not observed any increase in the Fe M$_{2,3}$ shift between the two samples, $\alpha$-Fe$_2$O$_3$/Al and $\alpha$-Fe$_2$O$_3$/parylene. Therefore, we deduced that either (i) the redox reaction is only partially taking place (reduction of $\alpha$-Fe$_2$O$_3$ by Al) in the time interval investigated or (ii) the reaction is maybe proceeding only near the interface, limited by the mass transport. According to the study from Shimojo et al.[21], the breaking of Fe-O bond and consequent formation of Al-O bonds at the interface would occur within few picoseconds (< 5 ps). Hence, the latter interpretation (reaction occurring only near the interface) is

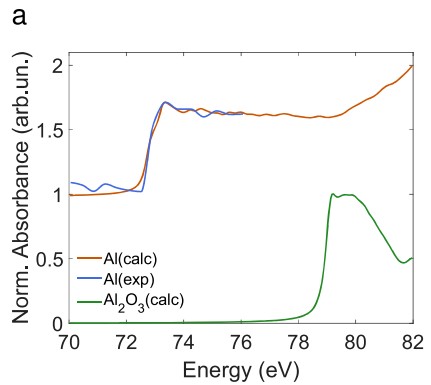

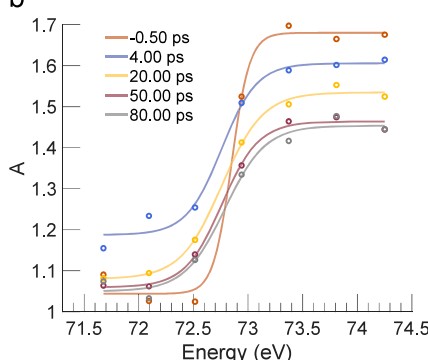

**Fig. 6 | Absorption spectra of Al L$_{2,3}$-edge. a** Simulations of the static absorption spectra of the Al L$_{2,3}$-edge for Al and $\alpha$-Al$_2$O$_3$ along with the measured static absorption spectrum of Al. **b** Absorption spectra of Al L$_{2,3}$-edge measured for the $\alpha$-Fe$_2$O$_3$/Al sample at five selected pump-probe delays. The flexible sigmoid function used for the fits in (**b**) is reported in the Supplementary Information, Eq. 11. Source data are provided as a Source Data file.

reasonable and the lack of spectroscopic signatures may be attributed to an insufficient contrast to see a relevant difference in presence of the aluminium.

### Transient absorption spectra at the Al L$_{2,3}$ edge

Concerning the Al L$_{2,3}$ absorption edge, simulations of the static absorption spectra of the L$_{2,3}$ edge for Al and $\alpha$-Al$_2$O$_3$ are reported in Fig. 6a. The absorption spectrum of $\alpha$-Al$_2$O$_3$ is predicted to be blue shifted by ~ 6 eV with respect to the spectrum of Al. In Fig. 6b, absorption spectra are reported for five selected delays in order to span the time window covered by the pump-probe measurements. A coherent blue shift of the absorption traces with time can be observed. Comparing the simulated spectra with the experimental time-dependent traces, we have made the following considerations: (i) according to Shimojo et al.[21], Al-O bonds form at the interface with $\alpha$-Fe$_2$O$_3$, but the aluminium layer is 100 nm thick, so the bond formation process affects only a small fraction of the Al atoms in the overall volume probed by the FEL beam, resulting in a blue-shift which is detectable but not dramatic as the one predicted in Fig. 6; (ii) the isochoric heating leads to hydrodynamic expansion and consequently to complete ablation of the sample within nanoseconds. On this time scale, the species migrate and interdiffuse between the two materials to a certain extent, due to the ultrafast raise of the temperature and the consequent strain that leads to the melting/disruption of the solids[14]. The expansion and the transition to the gas phase cause a drop of the density in the volume of sample probed by the FEL in the tens of picoseconds timescale, which, in turn, reduces the TA signal; (iii) the pressure in the experimental chamber was 10$^{-6}$ mbar and gaseous species may have probably been withdrawn from the reaction system, preventing the further proceeding of the reaction; (iv) the time window observed with the pump-probe is a hundred or few hundreds of ps and that may be not sufficient to detect the full progress of the reaction occurring.

For future investigations, the samples could be modified to avoid the problem of withdrawal of the gaseous species. Additional layers, transparent to the laser pump, may operate as a confinement for the sample; such layers would prevent the gases produced by the laser pump and by the heat released in the first the redox processes from escaping as a result of the low pressure in the experiment chamber. Another option would be that of using hard X-rays to probe the process at deeper core levels, which, on one hand, would not require working in vacuum, and, on the other hand, would offer the benefit of achieving a significantly higher penetration depth. This would allow to work with more complex sample structure, *e.g.* multilayers, which, in turn, would maximize the number of interfaces in the probed volume giving more contrast to observe changes associated with the redox

reaction. Further information could be gained by introducing an electric field to hinder/favor the electron transfer from aluminium to hematite and observe how such field would influence the transient absorption traces.

## Methods
### Samples
The samples investigated in this experiment were customized self-standing nanolayered thin filters, with a large area, approximately 20 mm diameter. The samples were provided by Luxel Corporation and include: (i) a nanometric $\alpha$-Fe$_2$O$_3$ layer deposited on an Al substrate for studying the thermite process, (ii) a nanometric $\alpha$-Fe$_2$O$_3$ layer (nominally 20 nm) deposited on a Parylene-C substrate (nominally 100 nm) and (iii) an Al sub-micrometric foil (100 nm) as a reference for the two independent species (Al and $\alpha$-Fe$_2$O$_3$). XAS measurements at the Fe M$_{2,3}$ edge and Al L$_{2,3}$ edge were conducted in separate experimental runs using different thermite samples. The obtained absorption levels (Fig. 2a,c) are consistent with a double-layer foil composed of 20 nm $\alpha$-Fe$_2$O$_3$ and 50 nm Al for the Al L$_{2,3}$ edge measurements, and a double-layer foil of 45 nm $\alpha$-Fe$_2$O$_3$ and 100 nm Al for the Fe M$_{2,3}$ edge measurements (see Supporting Information).

The samples of $\alpha$-Fe$_2$O$_3$/Al employed for the investigation on Fe M$_{2,3}$ edge and Al L$_{2,3}$ edge were different ones with the same nominal thickness. However, the actual thickness of the layers of $\alpha$-Fe$_2$O$_3$ and Al were different in the two samples.

### Pump-probe setup
The experiment was performed at the EIS-TIMEX end station[22] of the FERMI FEL[23] in Trieste, Italy (Fig. 1). TIMEX is an instrument devised for performing destructive pump-probe measurements to investigate systems in extreme conditions. The sample is rastered between each pump-probe measurement to always probe a fresh portion of the sample, which is crucial to study irreversible processes such as thermite reaction. FERMI is a seeded high-gain harmonic generation (HGHG) FEL[24]. The FERMI Ti:Sa femtosecond laser facility generates and shapes the seed pulses employed in the HGHG free-electron lasing process through a customized optical parametric amplifier (OPA)[25]. A portion of the fundamental emission at 785 nm from the Ti:Sa laser is delivered to the experimental hall and it is the standard laser used in pump-probe experiments[26], which was employed in this experiment. Such design provides a jitter of just few femtoseconds[27]. Hence, the aforementioned factors offer an experimental environment which is robust and well-suited for the present investigation. The duration of the pump pulse was about 80 fs FWHM, the spot size on the sample was 34 × 34 $\mu$m$^2$ FWHM and the intensity of the laser was set to 25 $\mu$J in order to reach a fluence of ~0.9 J·cm$^{-2}$ on the

sample. Such pump provokes irreversible damage on the sample (ablation) at each shot. Hence, the measurement is carried out in raster scan, single-shot mode, i.e. moving on a fresh region of the sample after each pump shot. We used the radiation produced by the FERMI FEL in the range of photon energy from 54 to 58 eV for probing at the Fe $M_{2,3}$ edge and in the range 73 to 76 eV for probing at the Al $L_{2,3}$ edge. The FEL pulse duration was 60 fs FWHM, and the intensity was 3 $\mu$J estimated by using a calibrated ionization chamber filled with $N_2$ and located along the beam transport. Spot size of the FEL was 16 $\times$ 16 $\mu m^2$ FWHM and the fluence at the sample was ~60$\mu$J $\cdot$ cm$^{-2}$ (no damage observed by the probe beam). The spectral broadening factor (i.e. $\Delta\lambda/\lambda$) for the emission of FERMI is of the order of $10^{-3}$ or $10^{-4}$ and no monochromators are used along the beam transport. The spectrum of the FEL along the beam transport is monitored by an EUV spectrometer, PRESTO[28]. The detector at TIMEX is a second EUV spectrometer, WEST[29].

### Static Fe $M_{2,3}$ edge calculations on FeO and $\alpha$-Fe$_2$O$_3$

The OCEAN code was used to simulate static Fe $M_{2,3}$ edge of two iron oxides with different Fe oxidation states (Fe$^{2+}$ for FeO and Fe$^{3+}$ for $\alpha$-Fe$_2$O$_3$)[30,31]. DFT+U calculations were performed with the Quantum ESPRESSO package using pseudopotentials with the Perdew-Burke-Ernzerhof generalized gradient approximation exchange-correlation functionals[32,33]. A cutoff energy of 240 Ry was used for the plane-wave basis set truncation in the DFT calculation. The self-consistent field calculation was done on a 6 × 6 × 6 k-point grid while the screening calculation was done on a 2 × 2 × 2 k-point grid. The number of bands in the screening calculation is consistent between the different structures and the same energy range of unoccupied bands (100 eV) was used for screening calculations for both iron oxides. OCEAN does not calculate the absolute energies of core-level excitations and relative peak shifts were calculated as a sum of the pseudopotential-dependent binding energy, total Kohn-Sham potential, and one-half the screened Coulomb potential of the core-hole[34,35].

All calculated spectra used identical energy shift and broadening. The calculated stick spectra were broadened with a Gaussian function with the linewidth in full width at half maximum (FWHM) at 0.4 eV (E ≤ 54 eV) and 0.8 eV (E > 54 eV). The energy axis and intensity of the simulated $\alpha$-Fe$_2$O$_3$ spectrum were first matched to the experimental spectrum and the simulated FeO spectrum was aligned using the relative energy shifts obtained from the OCEAN calculations. The Fe $M_{2,3}$ edge exhibits a characteristic peak located at 4.4 eV and 2.1 eV prior to the main peak for $\alpha$-Fe$_2$O$_3$ and FeO, respectively, which is consistent with reported experimental results[9,36].

### Static Al $L_{2,3}$ edge calculations on Al and $\alpha$-Al$_2$O$_3$

The same computational code was used to obtain static Al $L_{2,3}$ edge of Al and $\alpha$-Al$_2$O$_3$. These calculations were performed with optimized norm-conserving Vanderbilt pseudopotentials generated using the ONCVPSP code modified for the OCEAN code[37]. A cutoff energy of 240 Ry was set in the DFT calculations. The self-consistent field calculation was done on a 16 × 16 × 16 k-point grid while the screening calculation was done on a 2 × 2 × 2 k-point grid. The number of bands in the screening calculation is consistent between the different structures and the same energy range of unoccupied bands (200 eV) was used for screening calculations. The aforementioned approach to calculate relative peak shifts was used.

All calculated spectra used identical energy shift and broadening. The calculated unbroadened spectra were broadened with a Lorentzian linewidth in FWHM increasing linearly from 0.01 eV at 70 eV to 0.3 eV at 82 eV and higher. And the spectra were convoluted with a Gaussian function with FWHM calculated from the resolving power of the WEST spectrometer to account for instrumental broadening. The energy axis and intensity of the simulated Al spectrum were first matched to the experimental spectrum and the simulated $\alpha$-Al$_2$O$_3$

spectrum was aligned using the relative energy shift obtained from the OCEAN calculation. The Al $L_{2,3}$ edge of $\alpha$-Al$_2$O$_3$ is blue-shifted compared to that of Al due to the change in the oxidation state from Al$^{3+}$ to Al and the simulated spectra match well with the reported results using electron energy-loss spectroscopy[38].

## Data availability

The raw experimental data generated in this study have been deposited in this public repository: https://doi.org/10.34965/i60382[39]. Source data are provided with this paper.

## Code availability

The code employed to analyze the experimental data is available in this public repository: https://doi.org/10.34965/i60382[39].

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

## Acknowledgements

FEL radiation generation team, photon analysis, delivery and reduction system (PADReS) team and seed laser team are acknowledged for their work and assistance during the experiment performed at FERMI. EPal has received funding for this project from the European Union's Horizon 2020 research and innovation program under the Marie Skłodowska-Curie Grant Agreement No. 860553. This project was partially supported by the Liquid Sunlight Alliance (the U.S. Department of Energy, Office of Science, Office of Basic Energy Sciences, Fuels from Sunlight Hub, under Award Number DE-SC0021266) (SKC). WL acknowledges support by the Korea Foundation for Advanced Studies. The computations presented here were conducted in the Resnick High Performance Computing Center, a facility supported by the Resnick Sustainability Institute at the California Institute of Technology. The authors would like to acknowledge the contribution of prof. F. De Groot for the fruitful discussions on X-rays and EUV spectroscopies.

## Author contributions

RM and CM devised the experiment, EPa, JSPC, EPr, FB, DdA, LF, DN and RM performed the pump-probe experiments, WL performed the computations and wrote the computational part in the manuscript under the supervision of SKC, DG optimized FEL source, GK set up and aligned the pump laser, MM and AS optimized the FEL beam transport, EPa and JSPC performed the data analysis, EPa, JSPC, EPr and RM wrote the manuscript, all the authors revised the manuscript.

## Competing interests

The authors declare no competing interests.
