## [Transparent Peer Review file · Nature Communications]

Time-resolved chemically-selective spectroscopic investigation of the redox reaction between hematite and aluminium

Corresponding Author: Dr Emiliano Principi

Version 0:

Reviewer comments:

Reviewer #1

(Remarks to the Author)

This work aims to use transient XUV absorption spectroscopy to measure the first steps of the hematite/Al thermite reaction. This is a well-posed question, and the use of transient XUV spectroscopy is innovative. However, I cannot recommend this work for publication, for the reasons below.

Most importantly, by the authors own admission they do not actually measure anything conclusively. Much of page 9 is a discussion of possible reasons why the experiment did not work – for example there is no clear oxidation state change observed in the Al. I respect the authors for their honesty here, but without any scientific conclusions there is not enough novel about the technique to warrant publication in Nature Communications. The Conclusions are written more optimistically, but the results did not back that up – we cannot say that they have applied tr-XAS to “successfully monitor the evolution of a chemical reaction in several samples” when they have potentially only measured sample heating and/or ablation.

I had many other concerns with the manuscript.

1. The introduction is far too broad, going into topics such as glucose oxidation, capacitors, etc, that have very little to do with their work besides the fact that they involve oxidation state changes. I have no idea what statements like “the stress caused by the electron transfer is the driving force behind the bond breaking...” : the word “stress” has a specific meaning that has nothing to do with molecular electron transfer.

2. On page 2 there is the statement “The energy released by the process is higher [with nanoscaling]”. This seems suspect, though I could be wrong here: clearly the rate of the reaction will change based on the higher surface contact, but the final energy released is just thermodynamics and shouldn't care whether you reacted two small samples or one big sample.

3. The results and discussion section was very confusing: I had a very hard time figuring out whether they were measuring the dynamics of pure-hematite, pure-Al, or the bilayer film. The authors go back and forth between these samples, instead of clearly presenting one, then the next (subheadings would have been valuable here)

4. Figure 2 is confusing for several reasons. 2A shows energies of ~71.5 and ~73.3 as the selected energies for kinetics in 2C. However the labels on 2C are “72.09” and “73.37”. Is the “72.09” label wrong, or is the yellow circle at 71.5 wrong? In 2A, the data points are connected by some kind of spline fit – but there is a mysterious hump around 71.2 that seems totally disconnected from the data points. Where does that come from? Finally, it would be much easier for the reader if the scales in 2C and 2D are identical, so they can see whether the rise in 2C matches the rise in 2D. As plotted, the X axes are different in the two subplots.

5. I'm not convinced that the 116 ps and 40 ps components of the Fe pre/post edge are actually different. The data appears superimposable within the s/n of the measurement.

6. Figure 3 presents contour plots and spectral slices as if the data is continuous (for example with a broadband HHG source), but the actual data is very sparse (just 6-7 energies if it's the same spacing as in Figures 2A/2B). There's nothing wrong with measuring at just those energies (and it makes sense given their FEL source), but it is not honest to smear the

data out like this with some interpolation/spline and make it look so continuous. I'll come back to this point in the zero-crossing analysis. As a very minor note, the light grey for the static spectrum of Fe₂O₃ was very faint and hard to see even on my good laser printer.

7. In Figure 4, which forms the basis for their discussion of the time-evolution of the zero crossing, much of their analysis depends on whatever spline fit they use between their data points. They might get significantly different results if they had connected the data points with straight lines, or with a smoother spline, so I don't know how we can trust the values they got. There is also some kind of fit lines in 4D that are totally unexplained.

8. A major point: there are just too many possible things happening in the sample for them to form any conclusions about the chemical reaction. The Fe₂O₃ will show the initial charge transfer, then polaron formation, then potentially reduction to FeO or even to Fe. I don't see how this experiment has any hope of isolating the thermite chemical reaction given that they can't trigger heating of the Al without also photoexciting the hematite.

9. On that note, there is no estimate given for how much of the pump pulse is absorbed by the hematite vs by the Al. That would help the reader to figure out whether the majority of the signal they see in the Fe edge is caused by charge transfer to the Al, or whether they are just seeing hematite photophysics with a small perturbation from the Al.

10. Minor question: Al must have some reflectivity at 785 nm. Is the stronger signal for hematite/Al compared to pure hematite just because the reflection off the Al in hematite/Al results in higher absorption in the hematite? i.e. suppose for pure hematite, 50% of the pump is absorbed. For hematite/Al, is 50% absorbed in the hematite on the first pass, then 10% is absorbed in the Al and the other 40% reflects back to the hematite and excites it again (for an additional 20% excitation)?

In summary, this work is not ready for publication. Much more care needs to be put into the presentation of the manuscript, and I suspect that further experiments will be needed to come to any firm conclusions about the chemical reaction they want to study.

Reviewer #2

(Remarks to the Author)

The manuscript by Paltanin et al. reports the extension of the pump-probe technique to the study of heterogeneous redox chemical reactions on ultrafast time scales, harnessing the element-selectivity of the extreme ultraviolet (EUV) probe to separately investigate the reaction partners of the α -Fe₂O₃/Al thermite process.

Overall, the work shows data of the highest quality and presents a significant novelty in the X-ray community. The experiments were systematically performed at multiple time scales (fs-ps), at both Fe M_{2,3}-edge and Al L_{2,3}-edge and for both the sample undergoing the thermite reaction and for a reference sample (hematite on parylene) undergoing only photoinduced dynamics.

The requirements to trigger the chemical process, i.e. using a high pump intensity to induce a sufficient temperature change in the aluminum, are experimentally met. Furthermore, performing the measurement in destructive single-shot detection mode is the right approach for the investigation of this irreversible redox reaction, at least in the fs-ps time scales. The reproducibility of the results was guaranteed by performing raster scans along the sample surface. Systematic checks of the sample quality were performed at each sample position of the raster scan. The samples, experimental conditions and data analysis procedures are described in details, following the best practices of the community and allowing replicating the measurements.

The results show the feasibility of such an experiment, even at large scale facilities like free-electron lasers (FELs). The proposed interpretation is supported by the measured data, which, however, could provide even stronger conclusions with a deeper analysis of the available data.

This study is of a high significance for the broad chemistry, biochemistry and material science community since it represents a first step towards measuring the earliest events in bimolecular non-light driven redox processes. To our knowledge, no studies were so far reported for this kind of investigations in the EUV to soft X-ray regime.

Despite the significance of this study, the element-selectivity of the investigation strategy, which probes both the Fe and Al reaction partners, is not fully exploited to provide insights about the chemical reaction. In fact, most of the discussion is focused on the interpretation of the photoinduced polaron formation, which is already known in the literature and which does not correspond to the expected outcome of the thermite reaction. In order to consider the manuscript as suitable for publication in Nature Comm, the authors should deepen the interpretation of the available results by:

- (i) stressing the differences observed at the Fe M_{2,3}-edge for the Fe₂O₃/Al and Fe₂O₃/Parylene samples. This would help isolating the intrinsic time scales of the first reaction steps of the thermite reaction.
- (ii) providing an interpretation for the changes at the Al L_{2,3}-edge, possibly with the support of simulations.

Further, the authors should take into account the following comments:

1. In p.5, the fit of the Fe time trace at 56.49 eV misses 8 data points over 12 in the time range 0-1.2 ps. The authors should consider using an additional time constant to capture the ultrafast dynamics during the first 100 fs. This would be consistent with the time scale reported in the reference cited in the manuscript for polaron formation (90 fs) [Carneiro et al. Nature Materials, 16 (8):819–825, 2017]. In absence of significant improvements, the fit with 3 time constants could be included in the Supplementary information to further validate the biexponential fit.
2. In Figure 4a,b and Figure 4d the authors respectively use 0.3 ps and 0.1 ps as a reference for computing the zero-crossing shift. They should consider making a consistent choice for the reference time delay.

3. The authors should use the data reported in Figure 4d to show the time scales of the pump-probe experiment performed on the Fe₂O₃/Parylene sample and comment on the similarities/differences with respect to the Fe₂O₃/Al sample. Showing differences in the time scales of the dynamics would be the strongest proof that the proposed experiment is able to track the first steps of the thermite reaction, even if the signal is partially covered by the photodynamics triggered by the two-photon absorption process of hematite.
4. Figure 3(right) shows a pump-probe dynamics in the 1-50 ps time scale at the Al L_{2,3}-edge, with a ~0.3 eV shift of the zero-crossing point which is completely overlooked throughout the entire manuscript. The authors should consider computing the absorption spectrum of the Al and Al₂O₃ species to verify if the changes observed in their data agree with the formation of the reaction product, similarly to what is shown in Figure 4c for Fe₂O₃ and FeO. They should also show the simulated spectrum of the Fe₃O₄, which in the cited literature (ref. 3) is reported to be produced during the first steps of the thermite reaction.
5. In p.9 the interpretation reported in the following sentences is questionable: "According to the study from Shimojo et al.,³⁴ the breaking of Fe-O bond and consequent formation of Al-O bonds at the interface would occur within few picoseconds (< 5 ps). Hence, the latter interpretation (reaction occurring only near the interface) is reasonable and the lack of spectroscopic signatures may be attributed to an insufficient contrast to see a relevant difference in presence of the aluminium.". The authors see a strong response at the Al L_{2,3} edge (Figure 3b) with a significant red-shift of the zero-crossing point in the 2-20 ps time scale. This is also confirmed by the presence of a first time component in the fit shown in Figure 2b which is in good agreement with the time scales reported in reference 34. The message of the manuscript could be significantly strengthened by providing a deeper explanation of the changes observed at the Al edge, possibly performing simulations for the Al and Al₂O₃ systems.
6. The authors claim that "Highly non-linear electronic effects, e.g. tunneling ionization, multiphoton and avalanche ionization, caused by the pump pulse could be excluded, since they become dominant for peak powers higher than 1015 W/cm²,³³ while in this experiment the pump peak power was ~ 1013 W/cm²". While we agree that tunnelling is not occurring at this peak intensities, multiphoton and avalanche ionization can be significant. It would help to see the results of the time-resolved measurements performed at lower peak intensities where these effects can be ruled out.

Reviewer #3

(Remarks to the Author)

Version 1:

Reviewer comments:

Reviewer #1

(Remarks to the Author)

I continue to believe that the data presentation and explanation is not up to the standards of the field. Some specifics:

- In Figure 2A,C, the Y axis is "A(a.u.)". This should be actual absorbance units – otherwise the reader cannot judge the magnitude of the transient signal vs the ground state. Later they report ΔA values from +.65 to -1.03 (in the z scale of Figure 3-Left). Is that a real number, or some arbitrary scaling? Why not just report the actual A and ΔA (which is a rough measure of how much they have perturbed the sample)?
- Figure 2B,D reports ΔT . Equation 1 defines this term, but I've always seen this as $\Delta T/T$. Calling it ΔT will confuse the reader. And is there some scaling here? I would be surprised if the transmission really changed by +0.6 to -0.4 – that would be a huge change to the XUV transmission (i.e increasing or decreasing by ~50%)!
- On that note, why do they use units of ΔT in Figure 2 and ΔA in Figure 3? That is very confusing to the reader, since these are generally of opposite sign (positive ΔT is negative ΔA)
- I honestly do not understand what is being plotted in with the "0.2 ps" lines in Figure 3. These are not slices of the transient absorption, as they do not match either the contour plot or the general shape of the slices in Figure 4. For example, for the Al transient, the ΔA is positive at all times after t_0 below about 72.7 eV, and negative above 72.7 eV. But the green "0.2 ps" line is has the opposite signs (and with no zero line to show where $\Delta A=0$). This line also doesn't make sense as a reconstructed spectrum of what the Absorption spectrum would be at 0.2 ps. If it were that, then in the Fe spectrum there would be an isosbestic point at about 56.2 eV where $\Delta A=0$.
- Besides these substantive comments, it's silly to have four significant figures in the z scales of Figure 3 – that forces the font to be tiny which will be unreadable in the journal page.

It is still not clear when they are measuring pure Al or pure hematite, or the hematite/Al bilayer. I think that everything before Section 3.3 is pure-Al and pure-hematite, but the figure legends and the text don't actually say that.

I do appreciate the addition of Figure 4, which nicely shows the spectral evolution. Again, ΔA should be real and not arbitrary units, since they surely have the actual values.

I appreciate the discussion in the response to the reviewers about the hematite vs aluminum pump absorption (Point 9). However the text in the main paper is still not clear whether two-photon absorption is important – in section 3.2, the authors write "in the case of our experiment TPA is most likely the dominant mechanism of excitation". Does that mean that the hematite is absorbing a large percentage of the incident photons, or are they claiming it is negligible? Given that Figures 2-4

show pure-hematite TA (I think), the 2PA does seem to be important and should be considered when they discuss the bilayer.

As a general comment, I am not sure what the “general interest” criteria is for Nature Communications. I do believe that once the data is presented properly, this will be a good demonstration of the capabilities of the ELETTRA source. In my mind this would be an appropriate paper for a well-regarded specialty journal like J Phys Chem or Phys Chem Chem Phys, but I don't see the new scientific knowledge that would make it publishable in Chem Sci or JACS.

(Remarks on code availability)

Reviewer #2

(Remarks to the Author)

The manuscript has significantly improved thanks to the introduction of new figures and a better arrangement of the content with subheadings. The authors convincingly addressed all previous comments and the manuscript is now suitable for publication in Nat Comm.

As a final minor note, the clarity of the discussion would benefit from a consistent choice between absorption and transmission signals throughout the entire text. Figure 2b,2c and the related discussion are the only parts of the text referring to transient transmission, which has opposite sign with respect to transient absorption, making the text harder to follow.

(Remarks on code availability)

Reviewer #3

(Remarks to the Author)

(Remarks on code availability)

Version 2:

Reviewer comments:

Reviewer #1

(Remarks to the Author)

I appreciate the changes to the figure axes and the clarification of the TPA. I have two final factual concerns about the paper, where I think they made actual mistakes (and this is not just me having stylistic concerns)

First, I think there is an error in their calculation of the absorbance in Figure 2A/C. This is supposedly a film with 20 nm hematite on 100 nm Al. In Figure 2C, the absorbance axis goes from 2.2 at the pre-edge to 3.8 at the peak of the resonant signal. This gives a resonant intensity of 1.6. Previous work from the Leone group on hematite (<https://pubs.acs.org/doi/10.1021/jz401997d>) had resonant intensity of ~0.6 for 14 nm films. A 20 nm film should therefore have a resonant intensity of ~0.9, about half of what this paper reports. (comparison to predictions using the CXRO calculator at https://henke.lbl.gov/optical_constants/filter2.html also support the current film as being much more absorptive than predicted given the 20 nm thickness)

Of course, perhaps the Leone paper measured their hematite improperly. The real telling thing here is that Figures 2A and 2C are not internally consistent for being from the same sample. In 2C, the absorbance at 72 eV is 1. All previous XUV spectra of hematite show that the absorbance at about 70 eV is approximately the same as the absorbance before the peak (i.e. the decreasing photoionization cross section at higher energy is approximately balanced by the tail of the M-edge peak and the presence of 3p photoionization). Therefore the absorbance at 70 eV of the Fe₂O₃/Al sample should be ~2.2, matching the A at 55 eV. The authors should go back and look at their absorbance calculations – either the Y scale on Figure 2A is wrong, or the scale on Figure 2C is wrong. And once they figure out which, they should double-check their DeltaA calculations for the other figures.

Second, there appears to be something wrong with the hematite calculated spectrum in Figure 5C. From Cushing's previous calculations and from experiments, it should have a well-resolved peak at ~54 eV (<https://pubs.acs.org/doi/10.1021/acs.jpcc.2c06548>), not the broad thing shown in the figure. Honestly the FeO spectrum in red looks exactly like what Fe₂O₃ should look like, except for the position. Cushing should take a look at this figure and make sure things didn't get swapped somehow.

Reviewer #2

(Remarks to the Author)

Having read the complete response from the authors, I believe the manuscript is now suitable for publication in Nature Communications.

Reviewer #3

(Remarks to the Author)

Version 3:

Reviewer comments:

Reviewer #1

(Remarks to the Author)

I appreciate the corrections, and am now happy for this paper to be published in Nat Comm.

Report for the reviewers

Reviewer #1

1. The introduction is far too broad, going into topics such as glucose oxidation, capacitors, etc, that have very little to do with their work besides the fact that they involve oxidation state changes. I have no idea what statements like “the stress caused by the electron transfer is the driving force behind the bond breaking...” : the word “stress” has a specific meaning that has nothing to do with molecular electron transfer.

We have changed the sentence of the manuscript:

Redox reactions are fundamental chemical reactions that are involved in most of the processes essential for life on our planet, such as cellular respiration and photosynthesis, or processes that have a significant impact on human life, such as combustion and yeast fermentation. Electron transfer is the first process occurring in a redox reaction and is the fastest step in the entire reaction, typically within hundreds of femtoseconds. The fast variation of electronic distribution caused by the electron transfer is the driving force behind the bond breaking and formation process and the subsequent rearrangement of the nuclei, which takes place in the picoseconds time scale and beyond.

2. On page 2 there is the statement “The energy released by the process is higher [with nanoscaling]”. This seems suspect, though I could be wrong here: clearly the rate of the reaction will change based on the higher surface contact, but the final energy released is just thermodynamics and shouldn't care whether you reacted two small samples or one big sample.

We have changed the sentence of the manuscript:

Nanoscaling has mainly two beneficial effects: i) ignition temperature is decreased because of the lower melting point of nanometer size particles and ii) the rate of energy release of the process is higher thanks to the faster reaction kinetics, due to a more efficient surface contact between the reaction partners that maximizes mass transport.

3. The results and discussion section was very confusing: I had a very hard time figuring out whether they were measuring the dynamics of pure-hematite, pure-Al, or the bilayer film. The authors go back and forth between these samples, instead of clearly presenting one, then the next (subheadings would have been valuable here)

The results section has been organized in a different way with subheading to guide the reader through the process.

4. Figure 2 is confusing for several reasons. 2A shows energies of ~71.5 and ~73.3 as the selected energies for kinetics in 2C. However the labels on 2C are “72.09” and “73.37”. Is the “72.09” label wrong, or is the yellow circle at 71.5 wrong? In 2A, the data points are connected by some kind of spline fit – but there is a mysterious hump around 71.2 that seems totally disconnected from the data points. Where does that come from? Finally, it

would be much easier for the reader if the scales in 2C and 2D are identical, so they can see whether the rise in 2C matches the rise in 2D. As plotted, the X axes are different in the two subplots.

We kindly thank the reviewer for noticing this mistake. Indeed, The yellow circle at 71.68 was wrong, there was a mistake in the figure that the author did not notice. The actual number of energy points measured for Al was larger than those displayed, but the authors decided to show only those close to the absorption edge. The interpolation is displayed with the unique purpose of serving as a guide for the eye.

The x axis for figures 2b and 2d have been uniformed.

5. I'm not convinced that the 116 ps and 40 ps components of the Fe pre/post edge are actually different. The data appears superimposable within the s/n of the measurement.

We thank the reviewer for his careful analysis. Thanks to his comment, we have reviewed the fit boundaries, finding coherent time constants for the two dynamics.

6. Figure 3 presents contour plots and spectral slices as if the data is continuous (for example with a broadband HHG source), but the actual data is very sparse (just 6-7 energies if it's the same spacing as in Figures 2A/2B). There's nothing wrong with measuring at just those energies (and it makes sense given their FEL source), but it is not honest to smear the data out like this with some interpolation/spline and make it look so continuous. I'll come back to this point in the zero-crossing analysis. As a very minor note, the light gray for the static spectrum of Fe₂O₃ was very faint and hard to see even on my good laser printer.

We understand the concern of the reviewer, although it is clearly stated which energies are measured and that the aim of the 2D maps is that of providing an overall trend of the experimental observations, in line with the TA spectroscopy community.

To provide another perspective, an additional figure has been added to show transient absorption spectra at different pump-probe delays along with the sigmoid fits.

7. In Figure 4, which forms the basis for their discussion of the time-evolution of the zero crossing, much of their analysis depends on whatever spline fit they use between their data points. They might get significantly different results if they had connected the data points with straight lines, or with a smoother spline, so I don't know how we can trust the values they got. There is also some kind of fit lines in 4D that are totally unexplained.

In figure 5a and 5b, the points are fitted with a flexible sigmoid function weighted on the reciprocal of the square error, and the function is reported in the supporting info. We believe that the use of a sigmoid function is well-justified for fitting M and L edges and it is probably one of the best options to work with in our case, because it seizes the steep shape of absorption edges, while providing the flexibility of accounting for smearing and shifting expected to be observed in transient phenomena. We tried to use a different function, a simple interpolation of the energy points fitted across the absorption edge. The values of the individual shifts of the zero-crossing over as a function of pump-probe delays vary, but there is still a similar difference between the two samples (Fe₂O₃/Al and Fe₂O₃/Parylene).

Anyhow, we believe that the choice of the weighted sigmoid function is a more rigorous approach. Concerning Fig. 5 d, the lines serve merely as guides for the eyes for highlighting the different dynamics of the zero-crossing points over the pump-probe delay for the two samples compared.

8. A major point: there are just too many possible things happening in the sample for them to form any conclusions about the chemical reaction. The Fe_2O_3 will show the initial charge transfer, then polaron formation, then potentially reduction to FeO or even to Fe . I don't see how this experiment has any hope of isolating the thermite chemical reaction given that they can't trigger heating of the Al without also photoexciting the hematite.

The referee is right. Visible photoexcitation and subsequent EUV probing unavoidably come with the complication of observing several phenomena other than the chemical reaction, especially in solid-state heterogeneous reactions. A broad study of consolidated literature in the fields of thermite reactions, of the effects of ultrafast laser pulses in solids and EUV spectroscopy, was at the roots of the interpretation of the results provided by authors. Ultrafast spectroscopy is a technique that could attempt to temporarily resolve both the reaction and the photophysical phenomena.

Regardless, the authors relied on studies on thermite reactions where a laser was employed to trigger the reaction (such as ref 5, where the reaction is investigated with a 337 nm laser).

9. On that note, there is no estimate given for how much of the pump pulse is absorbed by the hematite vs by the Al . That would help the reader to figure out whether the majority of the signal they see in the Fe edge is caused by charge transfer to the Al , or whether they are just seeing hematite photophysics with a small perturbation from the Al .

We have performed absorption measurements in the visible spectral range on the samples that we have employed in the experiment, namely hematite on parylene and parylene. This has allowed us to extract an absorption spectra of hematite in the visible. At the pump wavelength (highlighted in the figure with the dashed gray line), the absorbed light by hematite is less than 2%, hence only a small fraction of light is absorbed by the hematite layer, while most of the light passes through it.

10. Minor question: Al must have some reflectivity at 785 nm. Is the stronger signal for hematite/Al compared to pure hematite just because the reflection off the Al in hematite/Al results in higher absorption in the hematite? i.e. suppose for pure hematite, 50% of the pump is absorbed. For hematite/Al, is 50% absorbed in the hematite on the first pass, then 10% is absorbed in the Al and the other 40% reflects back to the hematite and excites it again (for an additional 20% excitation)?

We expect aluminum (Al) to have a high reflectivity of about 85% at 785 nm. However, we measured a reflectance of 60%, which we believe is due to scattering from the free-standing film. In point 9, we demonstrate that hematite has very weak absorption at these wavelengths (smaller than 2% for one-photon absorption). Therefore, we expect the reflectivity of the hematite + aluminum sample to be similar to that of the free-standing aluminum film. Furthermore, given the scattering that we cannot rigorously quantify, it is reasonable to think that most of the heating/absorption would take place in the aluminium film.

Additionally, considering the two-photon absorption mechanism for polaron formation, if the photons reflected from the Al substrate increased the number of photogenerated polarons, we would expect a larger blue-shift of the Fe M_{2,3}-edge with Al as a substrate. This is because the blue-shift is associated with polaron formation, hence the greater the number of polaron generated, the larger the blue-shift expected to be observed. However, since we observe the opposite trend, i.e. a smaller blue-shift, we conclude that the back-reflected photons do not generate a significant number of two-photon absorption events.

Reviewer #2/3

In order to consider the manuscript as suitable for publication in Nature Comm, the authors should deepen the interpretation of the available results by:

- (i) stressing the differences observed at the Fe M_{2,3}-edge for the Fe₂O₃/Al and Fe₂O₃/Parylene samples. This would help isolating the intrinsic time scales of the first reaction steps of the thermite reaction.

(ii) providing an interpretation for the changes at the Al L_{2,3}-edge, possibly with the support of simulations.

We are grateful for the care and the effort that the reviewers have dedicated in the analysis of the paper. The suggestions made have been really enlightening and have led to a considerable amount of work, rewarded with a consistent improvement of the manuscript. The authors reviewed the manuscript in order to stress the differences of Fe M_{2,3}-edge in the Fe₂O₃/Al and Fe₂O₃/Parylene samples and make the discussion easier to follow. The authors have provided additional simulations to account for the changes at the Al L_{2,3}-edge.

Further, the authors should take into account the following comments:

1. In p.5, the fit of the Fe time trace at 56.49 eV misses 8 data points over 12 in the time range 0-1.2 ps. The authors should consider using an additional time constant to capture the ultrafast dynamics during the first 100 fs. This would be consistent with the time scale reported in the reference cited in the manuscript for polaron formation (90 fs) [Carneiro et al. Nature Materials, 16 (8):819–825, 2017]. In absence of significant improvements, the fit with 3 time constants could be included in the Supplementary information to further validate the biexponential fit.

We used two exponentials for fitting the dynamic at 56.49 eV. The best model, as discussed in the SI, is the following:

$$f(x) = \frac{1}{2} \left(1 + \operatorname{erf} \left(\frac{x}{\sigma \sqrt{2}} \right) \right) \times (A1 \exp(-x/\tau_1) + A2 \exp(-x/\tau_2)) - 0.21$$

The model for the fast part of the dynamics, takes into account the time resolution and two exponential decay for the equilibration between carriers and the lattice.

The biexponential model seems to work quite well. One possible explanation for the fast τ may be the formation of the polaron, which we see as a decay in terms of depletion of the conduction band electrons that results from the self-trapping of the carrier. In the paper from Carneiro et al. (Ref. 8 of the main paper), the polaron formation is a process reported to be in the order of 100 fs, which is similar to the fast τ that we have obtained from the fit (0.1 ± 0.06 ps).

2. In Figure 4a,b and Figure 4d the authors respectively use 0.3 ps and 0.1 ps as a reference for computing the zero-crossing shift. They should consider making a consistent choice for the reference time delay.

Fig. 4A and 4B have been adapted to make a consistent choice.

3. The authors should use the data reported in Figure 4d to show the time scales of the pump-probe experiment performed on the Fe₂O₃/Parylene sample and comment on the similarities/differences with respect to the Fe₂O₃/Al sample. Showing differences in the time scales of the dynamics would be the strongest proof that the proposed experiment is able to track the first steps of the thermite reaction, even if the signal is partially covered by the photodynamics triggered by the two-photon absorption process of hematite.

Our study primarily focuses on the transient variation of the shift of the zero-crossing point in the pump-probe signals. As the referees reported, by examining the zero-crossing point, we can capture the subtle dynamics that occur on ultrafast time scales, which are essential for understanding the initial stages of the thermite reaction. However, in Fe₂O₃/Al we are

observing two competing phenomena in our experiments: the formation of small polarons and the electron transfer from aluminum, each with its distinct time scale and dynamics. These phenomena are inherently coupled in the system and cannot be resolved individually. Furthermore, the evolution of the individual values of the zero-crossing for the two samples does not conform to a straightforward functional form, because the non-linear nature of the zero-crossing point evolution substantiates the complexity of the early-stage reaction dynamics, which our proposed experiment is capable of tracking. For this reason the authors chose to compare the magnitude of the shift in time of the zero-crossing point rather than fitting time constants for the two dynamics reported in Fig. 5 d (in the revised manuscript, old Fig. 4 is Fig. 5).

4. Figure 3(right) shows a pump-probe dynamics in the 1-50 ps time scale at the Al L_{2,3}-edge, with a ~0.3 eV shift of the zero-crossing point which is completely overlooked throughout the entire manuscript. The authors should consider computing the absorption spectrum of the Al and Al₂O₃ species to verify if the changes observed in their data agree with the formation of the reaction product, similarly to what is shown in Figure 4c for Fe₂O₃ and FeO. They should also show the simulated spectrum of the Fe₃O₄, which in the cited literature (ref. 3) is reported to be produced during the first steps of the thermite reaction.

Calculations at the Al L_{2,3} edge have been performed and compared to the experimental data (see manuscript). As reported below from experimental spectra, up to before 20 ps, there is a red-shift which is puzzling. From 20 ps onward, we observe a blue-shift (See the first plot reported below, which shows absorption spectra at different delays). For Al L edge, there is a strong blue shift predicted upon oxidation of Al to Al₂O₃ (see the second plot below). An interpretation for the experimental observation, which differs from the prediction, could be the following: the red-shift could be associated with electron transfer to Fe₂O₃ (from 0.05 ps to 4 ps) and the subsequent blue-shift could be associated with the formation of Al-O bonds at the interface (from 4ps to 80 ps).

The static spectrum of the Fe M_{2,3}-edge in Fe₃O₄ has been calculated and is reported in the third panel below, together with the simulated spectra of FeO and Fe₂O₃. It exhibits two main peaks as for the FeO spectrum, but it displays two main differences: i) the ratio of the the intensities of the peaks result different and ii) the position of the peaks for Fe₃O₄ are blue-shifted with respect to FeO, which is in line with the expected trend, i.e. the Fe M_{2,3}-edge red-shifts upon reduction.

5. In p.9 the interpretation reported in the following sentences is questionable: “According to the study from Shimojo et al.,³⁴ the breaking of Fe-O bond and consequent formation of Al-O bonds at the interface would occur within few picoseconds (< 5 ps). Hence, the latter interpretation (reaction occurring only near the interface) is reasonable and the lack of spectroscopic signatures may be attributed to an insufficient contrast to see a relevant difference in presence of the aluminium.”. The authors see a strong response at the Al L_{2,3} edge (Figure 3b) with a significant red-shift of the zero-crossing point in the 2-20 ps time scale. This is also confirmed by the presence of a first time component in the fit shown in

Figure 2b which is in good agreement with the time scales reported in reference 34. The message of the manuscript could be significantly strengthened by providing a deeper explanation of the changes observed at the Al edge, possibly performing simulations for the Al and Al₂O₃ systems.

We are grateful for the observation of the referees and thanks to their suggestion we performed simulations on the Al L_{2,3} edge. In light of the comparison with the simulated absorption spectra we have made additional considerations for the results of the Al L_{2,3} absorption edge, drawing some more conclusion on the electron transfer process and the possibility of formation of Al-O bond, addressing the concern raised by the referee.

6. The authors claim that “Highly non-linear electronic effects, e.g. tunneling ionization, multiphoton and avalanche ionization, caused by the pump pulse could be excluded, since they become dominant for peak powers higher than 10¹⁵ W/cm²,³³ while in this experiment the pump peak power was ~ 10¹³ W/cm²”. While we agree that tunnelling is not occurring at this peak intensities, multiphoton and avalanche ionization can be significant. It would help to see the results of the time-resolved measurements performed at lower peak intensities where these effects can be ruled out.

On the one hand, performing this experiment at the FERMI FEL facility has been a great opportunity and has allowed us to exploit a unique experimental setup which is robust and ideal for destructive pump-probe investigation. On the other hand, accessing a large-scale facility and being granted time slots is very difficult, therefore we could not perform additional measurements. For instance, there wasn't the possibility of performing a scan of the pump fluence. It would have been really interesting and it could have provided significant additional information, but we did not have this opportunity. Therefore, we just removed this misleading sentence.

Report for the referees

We thank the reviewers for their valuable feedback and for the effort they have put into helping us improve our work. We have made several revisions to the manuscript, and below we address each of the reviewers' comments point by point.

1) I continue to believe that the data presentation and explanation is not up to the standards of the field. Some specifics:

- In Figure 2A,C, the Y axis is "A(a.u.)". This should be actual absorbance units – otherwise the reader cannot judge the magnitude of the transient signal vs the ground state. Later they report ΔA values from +.65 to -1.03 (in the z scale of Figure 3-Left). Is that a real number, or some arbitrary scaling? Why not just report the actual A and ΔA (which is a rough measure of how much they have perturbed the sample)?

In Fig 2A and 2C the absorbance has been thoroughly calculated, as explained in the supporting info. A measure is performed without the sample for every measured photon energy to establish the correlation between the PRESTO and the WEST spectrometers in order to account for the transmission of the TIMEX beamline. The intensity before the sample measured with the PRESTO spectrometer is used as the I_0 and then corrected with the aforementioned correlation function, while I_1 is taken as the transmission of the unperturbed sample (without the effect induced by the pump beam). This is the actual transmission for each shot, and a reliable value is obtained through a binning procedure to account for statistical fluctuations. From the average value of transmission, we had calculated the absorbance. The absorbance is a unitless number, which depends as well on the thickness of the sample. In Fig. 2 we aim just to show the shape of the absorption edge measured at the FEL to demonstrate that it is comparable with the ones measured at the synchrotron and to show the disposition of the photon energies that we have measured. Nonetheless, we have changed the figures by adding the absorbance values. The most common choice we found in the literature is reporting absorbance with absorbance units or arbitrary units. We decided to use just A and ΔA without adding (A.U.) for absorbance units in order to avoid issue related with scaling factors or ambiguity. This choice has been adopted in several works on the literature [1,2,3,4]. Figures 5c and 6a are computed simulations of static absorption spectra. In this case the computed value of absorbance was actually arbitrary and it has been normalized using the maximum of the A value in the experimental static spectra of Fe M_{2,3} edge and Al L_{2,3} edge. Concerning the figures featuring ΔA , that is calculated as a difference of absorbance values at different delays with respect to the absorbance of the unperturbed sample, therefore those are as well actual absorbance units, there is no scaling factors. At a first glance, the magnitude of the variation of the absorbance values may seem unreasonable, but it is resulting from the excitation of the sample with an high intensity laser pump. The sample is being driven far from its equilibrium condition and it undergoes an irreversible damage at each pump shot. Such strong excitations can be regarded as a well-justified cause for the variation of absorbance observed in this experiment.

[1] Pascarelli, Sakura, et al. "Energy-dispersive absorption spectroscopy for hard-X-ray micro-XAS applications." *Journal of synchrotron radiation* 13.5 (2006): 351-358.

[2] Summers, Adam M., et al. "Realizing attosecond core-level X-ray spectroscopy for the investigation of condensed matter systems." *Ultrafast Science* 3 (2023): 0004.

[3] Brittain, Harry G., ed. *Profiles of drug substances, excipients, and related methodology*. Academic press, 2020.

[4] Reisberg, P. I., and J. S. Olson. "Equilibrium binding of alkyl isocyanides to human hemoglobin." *Journal of Biological Chemistry* 255.9 (1980): 4144-4150.

2) Figure 2B,D reports ΔT . Equation 1 defines this term, but I've always seen this as $\Delta T/T$. Calling it ΔT will confuse the reader. And is there some scaling here? I would be surprised if the transmission really changed by +0.6 to -0.4 – that would be a huge change to the XUV transmission (i.e increasing or decreasing by ~50%)!

We understand the concern of the referee. Fig. 2 has been modified as it will be explained in point 3 to address the minor note of Reviewers 2 and 3 as well. As for the magnitude of the variation in the transmission, this issue has been commented at the end of point 1.

3) On that note, why do they use units of ΔT in Figure 2 and ΔA in Figure 3? That is very confusing to the reader, since these are generally of opposite sign (positive ΔT is negative ΔA)

All the figures in the paper feature absorption spectra and their evolution in time upon perturbation from the pump pulses. To be coherent with the convention adopted in the spectroscopy community, we reported all the spectra in a format absorbance vs photon energy.

Figures 2b and 2d actually are dynamics for a single energy. The information provided from these dynamics is the temporal evolution and reporting the plot with transmission or absorbance is not substantially different from the perspective of the temporal evolution of the system. Nevertheless, we have reported the dynamics in Figures 2b and 2d using absorbance instead of transmission to be coherent throughout the paper, as suggested by the referees.

4) I honestly do not understand what is being plotted in with the "0.2 ps" lines in Figure 3. These are not slices of the transient absorption, as they do not match either the contour plot or the general shape of the slices in Figure 4. For example, for the Al transient, the ΔA is positive at all times after t_0 below about 72.7 eV, and negative above 72.7 eV. But the green "0.2 ps" line is has the opposite signs (and with no zero line to show where $\Delta A=0$). This line also doesn't make sense as a reconstructed spectrum of what the Absorption spectrum would be at 0.2 ps. If it were that, then in the Fe spectrum there would be an isosbestic point at about 56.2 eV where $\Delta A=0$. - Besides these substantive comments, it's silly to have four significant figures in the z scales of Figure 3 – that forces the font to be tiny which will be unreadable in the journal page.

The bottom panels in fig.3 are absorption spectra at the delay $t=0$ and $t=0.2$ ps. The y axis displays A, differently from the 2D maps above in Fig. 3 which displays ΔA . The aim of the panels at the bottom is that of helping the reader to understand the trend of ΔA over time by visualizing the evolution of the absorption spectra with increasing pump-probe delay. Nonetheless, since it is not crucial information, we decided to remove the panels to improve the readability of the figure, as suggested by the referee.

It is still not clear when they are measuring pure Al or pure hematite, or the hematite/Al bilayer. I think that everything before Section 3.3 is pure-Al and pure-hematite, but the figure legends and

the text don't actually say that.

In order to address this concern, we have added a specification of the measured sample at the beginning of the results section and we have specified the sample being measured in all the figures, highlighting it in the caption where we believed it could be necessary.

I do appreciate the addition of Figure 4, which nicely shows the spectral evolution. Again, ΔA should be real and not arbitrary units, since they surely have the actual values.

We addressed this concern in point 1.

I appreciate the discussion in the response to the reviewers about the hematite vs aluminum pump absorption (Point 9). However the text in the main paper is still not clear whether two-photon absorption is important – in section 3.2, the authors write “in the case of our experiment TPA is most likely the dominant mechanism of excitation”. Does that mean that the hematite is absorbing a large percentage of the incident photons, or are they claiming it is negligible? Given that Figures 2-4 show pure-hematite TA (I think), the 2PA does seem to be important and should be considered when they discuss the bilayer.

Thank you for pointing out this issue. We realized that the original text was somewhat unclear, and we have simplified the presentation to make the mechanism more straightforward, as suggested:

“According to the literature, two-photon absorption (TPA) is the dominant mechanism for excitation in hematite for photon energies below its bandgap. Okazaki et al. demonstrated this by studying the excitation mechanism of a hematite photoanode in contact with a gold nanorod using transient absorption spectroscopy. Their findings showed that TPA must be considered to explain the dynamics of hematite’s transient absorption spectra when using an excitation wavelength of 800 nm. Additionally, good efficiencies for non-linear optical processes in the near-infrared region, such as at 1064 and 1340 nm, further support the occurrence of TPA in hematite at longer wavelengths relative to its visible bandgap”.

In our interpretation, hematite is absorbing a small fraction of the incident photons, but since we are employing a high fluence of the laser pump, a small fraction is absorbed by a TPA process.

As a general comment, I am not sure what the “general interest” criteria is for Nature Communications. I do believe that once the data is presented properly, this will be a good demonstration of the capabilities of the ELETTRA source. In my mind this would be an appropriate paper for a well-regarded specialty journal like J Phys Chem or Phys Chem Chem Phys, but I don't see the new scientific knowledge that would make it publishable in Chem Sci or JACS.

Reviewer #2 (Remarks to the Author):

The manuscript has significantly improved thanks to the introduction of new figures and a better arrangement of the content with subheadings. The authors convincingly addressed all previous comments and the manuscript is now suitable for publication in Nat Comm.

As a final minor note, the clarity of the discussion would benefit from a consistent choice between absorption and transmission signals throughout the entire text. Figure 2b,2c and the related

discussion are the only parts of the text referring to transient transmission, which has opposite sign with respect to transient absorption, making the text harder to follow.

We have adopted Figure 2b and 2d to make them coherent with the rest of the figures in the paper.

REPLY TO REVIEWER COMMENTS

Reviewer #1 (Remarks to the Author):

I appreciate the changes to the figure axes and the clarification of the TPA. I have two final factual concerns about the paper, where I think they made actual mistakes (and this is not just me having stylistic concerns).

First, I think there is an error in their calculation of the absorbance in Figure 2A/C. This is supposedly a film with 20 nm hematite on 100 nm Al. In Figure 2C, the absorbance axis goes from 2.2 at the pre-edge to 3.8 at the peak of the resonant signal. This gives a resonant intensity of 1.6. Previous work from the Leone group on hematite (<https://pubs.acs.org/doi/10.1021/jz401997d>) had resonant intensity of ~0.6 for 14 nm films. A 20 nm film should therefore have a resonant intensity of ~0.9, about half of what this paper reports. (comparison to predictions using the CXRO calculator at https://henke.lbl.gov/optical_constants/filter2.html also support the current film as being much more absorptive than predicted given the 20 nm thickness)

Of course, perhaps the Leone paper measured their hematite improperly. The real telling thing here is that Figures 2A and 2C are not internally consistent for being from the same sample. In 2C, the absorbance at 72 eV is 1. All previous XUV spectra of hematite show that the absorbance at about 70 eV is approximately the same as the absorbance before the peak (i.e. the decreasing photoionization cross section at higher energy is approximately balanced by the tail of the M-edge peak and the presence of 3p photoionization). Therefore the absorbance at 70 eV of the Fe₂O₃/Al sample should be ~2.2, matching the A at 55 eV. The authors should go back and look at their absorbance calculations – either the Y scale on Figure 2A is wrong, or the scale on Figure 2C is wrong. And once they figure out which, they should double-check their DeltaA calculations for the other figures.

REPLY: We are grateful to the referee for pointing out the discrepancies between the spectra in Figures 2a/2c and the nominal sample thickness. The XAS measurements were conducted in different experimental runs, as the FERMI FEL could not efficiently operate at both the Al L-edge and Fe M-edge during the same session. Moreover, the samples were destroyed after fs-laser exposure, requiring us to use different samples from distinct production lots.

Following referee #1's suggestion, we used the CXRO calculator and found that the actual thickness of the thermite samples differs from the nominal one. This is not uncommon, as the commercial production of submicrometric double layers can result in significant deviations in the final thickness. We have amended the text and added a new section in the Supplementary Information (see list of changes) to provide a realistic estimation of the sample thickness, compatible with the XAS spectra shown in Figures 2a and 2c.

However, we emphasize that the static spectra primarily serve as a reference for the time-resolved XAS measurements, which remain robust in their interpretation. Given the thorough data analysis outlined in the Supporting Information, we believe that the key conclusions of our work remain unaffected, as they are based on the relative temporal variation of absorption rather than its absolute value. That said, if deemed necessary, we are open to removing Figures 2a and 2c to prevent any potential confusion.

Second, there appears to be something wrong with the hematite calculated spectrum in Figure 5C. From Cushing's previous calculations and from experiments, it should have a well-resolved peak at ~54 eV (<https://pubs.acs.org/doi/10.1021/acs.jpcc.2c06548>), not the broad thing shown in the figure. Honestly the FeO spectrum in red looks exactly like what Fe₂O₃ should look like, except for the position. Cushing should take a look at this figure and make sure things didn't get swapped somehow.

REPLY: We appreciate the reviewer's feedback regarding the calculated spectrum of hematite. Due to the limited experimental data, which spans only energy points between 55 and 58 eV, the broadening applied to the

calculated spectrum in order to match the experimental data introduces some discrepancies compared to spectra from previous experiments and calculations. To address this, we have applied an alternative broadening method to be more consistent with energy dependent broadening in literature, which is detailed in the revised manuscript (see list of changes). This new approach improves the match between the calculated (blue) and experimental (black) spectra, including the pre-edge peak at ~54 eV. Additionally, we would like to emphasize that the same broadening method has been applied to the calculated spectrum of FeO (orange). This results in a better agreement with the spectral features observed in previous studies of the Fe $M_{2,3}$ edge for Fe²⁺-bearing compounds (van Aken, P. A., Styrsa, V. J., Liebscher, B., Woodland, A. B. & Redhammer, G. J. Microanalysis of Fe³⁺/ Σ Fe in oxide and silicate minerals by investigation of electron energy-loss near-edge structures (ELNES) at the Fe $M_{2,3}$ edge. *Phys Chem Min* **26**, 584–590 (1999)), as well as for Fe²⁺ states (Vura-Weis, J. *et al.* Femtosecond $M_{2,3}$ -Edge Spectroscopy of Transition-Metal Oxides: Photoinduced Oxidation State Change in α -Fe₂O₃. *J. Phys. Chem. Lett.* **4**, 3667–3671 (2013)).

Finally, we have carefully verified that the spectra were not inadvertently swapped. We appreciate the reviewer's vigilance in raising this possibility and confirm that all spectral assignments are correct.

Reviewer #2 (Remarks to the Author):

Having read the complete response from the authors, I believe the manuscript is now suitable for publication in Nature Communications.

REPLY: We sincerely thank the referee for recommending the publication of our manuscript.

Reviewer #3 (Remarks to the Author):

REPLY: We thank the referee for their valuable contributions in improving our manuscript.

LIST OF CHANGES

Page 2, “Materials” section

Previous sentence:

The samples were provided by Luxel Corporation, namely: i) α -Fe₂O₃ 20 nm deposited on Al 100 nm (nominal thickness) to study the thermite process and ii) α -Fe₂O₃ 20 nm deposited on Parylene-C 100 nm and iii) Al 100 nm to use as references for the two independent individual species (Al and α -Fe₂O₃).

New sentence:

The samples were provided by Luxel Corporation and include: (i) a nanometric α -Fe₂O₃ layer deposited on an Al substrate for studying the thermite process, (ii) a nanometric α -Fe₂O₃ layer (nominally 20 nm) deposited on a Parylene-C substrate (nominally 100 nm), and (iii) an Al sub-micrometric foil (nominally 100 nm) as a reference for the two independent species (Al and α -Fe₂O₃). XAS measurements at the Fe M_{2,3} edge and Al L_{2,3} edge were conducted in separate experimental runs using different thermite samples. The obtained absorption levels (Fig. 2a and 2c) are consistent with a double-layer foil composed of 20 nm α -Fe₂O₃ and 50 nm Al for the Al L_{2,3} edge measurements, and a double-layer foil of 45 nm α -Fe₂O₃ and 100 nm Al for the Fe M_{2,3} edge measurements (see Supplementary Information).

Page 4, “Materials” section

Previous sentence: **Static Fe M_{2,3} edge calculations on FeO and α -Fe₂O₃ [...]** All calculated spectra used identical energy shift and broadening. The calculated stick spectra were broadened with a Lorentzian function with the linewidth in full width at half maximum (FWHM) increasing linearly from 0.1 eV at 53 eV to 0.7 eV at 58 eV and higher. This energy-dependent broadening was used because higher-lying energy states have shorter core-hole lifetimes. And a Fano parameter of 3.5 was applied to the spectra due to the interference between two pathways sharing the same initial and final states which are $3p^63d^n \rightarrow 3p^63d^{n-1}\epsilon f$ direct transitions and $3p^63d^n \rightarrow 3p^63d^{n+1} M_{2,3}$ transitions followed by the autoionization to $3p^63d^{n-1}\epsilon f$ via super Coster-Kronig transition.²⁵ Finally, the spectra were convoluted with a Gaussian function with FWHM calculated from the resolving power of the WEST spectrometer to account for instrumental broadening. The energy axis and intensity of the simulated α -Fe₂O₃ spectrum were first matched to the experimental spectrum and the simulated FeO spectrum was aligned using the relative energy shifts obtained from the OCEAN calculations. The Fe M_{2,3} edge exhibits a characteristic peak located at **4.0 eV** and **2.3 eV** prior to the main peak for α -Fe₂O₃ and FeO, respectively, which is consistent with reported experimental results.^{7,26}

New sentence: **Static Fe M_{2,3} edge calculations on FeO and α -Fe₂O₃ [...]** All calculated spectra used identical energy shift and broadening. The calculated stick spectra were broadened with a Gaussian function with the linewidth in full width at half maximum (FWHM) at 0.4 eV ($E \leq 54$ eV) and 0.8 eV ($E > 54$ eV). The energy axis and intensity of the simulated α -Fe₂O₃ spectrum were first matched to the experimental spectrum and the simulated FeO spectrum was aligned using the relative energy shifts obtained from the OCEAN calculations. The Fe M_{2,3} edge exhibits a characteristic peak located at **4.4 eV** and **2.1 eV** prior to the main peak for α -Fe₂O₃ and FeO, respectively, which is consistent with reported experimental results.^{7,25}

Supplementary information

Changes: An additional section titled 'Sample Thickness Assessment' has been added to present the evaluation of the sample thicknesses using the CXRO calculator, as suggested by Reviewer #1.

New text: The thickness assessment of the double-layer thermite foils used in our XAS measurements in transmission geometry was performed using the CXRO calculator (https://henke.lbl.gov/optical_constants/filter2.html). In the calculations, densities of 5.26 g/cm³ and 2.70 g/cm³ were considered for α -Fe₂O₃ and aluminum, respectively. Oxidation of the free surfaces of the double foil due to atmospheric oxygen was neglected.

Figure 8 shows the theoretical absorption of α -Fe₂O₃ and aluminum thin foils across the Fe M_{2,3} edge and Al L_{2,3} edge for two different thickness combinations corresponding to the samples used in our experiments:
a) absorption of a double-layer consisting of 45 nm α -Fe₂O₃ and 100 nm Al;
b) absorption of a double-layer consisting of 20 nm α -Fe₂O₃ and 50 nm Al.

Figure 8 caption text: Theoretical absorption calculation across the Fe M_{2,3} edge and Al L_{2,3} edges for selected α -Fe₂O₃ and Al nanometric foils